# Three Distinct Reporter Systems of Hepatitis E Virus and Their Utility as Drug Screening Platforms

**DOI:** 10.3390/v15101989

**Published:** 2023-09-23

**Authors:** Putu Prathiwi Primadharsini, Shigeo Nagashima, Takashi Nishiyama, Hiroaki Okamoto

**Affiliations:** 1Division of Virology, Department of Infection and Immunity, Jichi Medical University School of Medicine, 3311-1 Yakushiji, Shimotsuke 329-0498, Tochigi, Japan; thiwik8@jichi.ac.jp (P.P.P.); shigeon@jichi.ac.jp (S.N.); 2Laboratory of Membrane Proteins, Research Division for Quantitative Life Sciences, Institute for Quantitative Biosciences, The University of Tokyo, 1-1-1 Yayoi, Bunkyo-ku, Tokyo 113-0032, Japan; tnishiyama@iqb.u-tokyo.ac.jp

**Keywords:** hepatitis E virus, virus life cycle, virus reporter system, GLuc, nanoKAZ, HiBiT, drug screening

## Abstract

The hepatitis E virus (HEV) is increasingly acknowledged as the primary cause of acute hepatitis. While most HEV infections are self-limiting, cases of chronic infection and fulminant hepatitis necessitate the administration of anti-HEV medications. However, there is a lack of specific antiviral drugs designed for HEV, and the currently available drug (ribavirin) has been associated with significant adverse effects. The development of innovative antiviral drugs involves targeting distinct steps within the viral life cycle: the early step (attachment and internalization), middle step (translation and RNA replication), and late step (virus particle formation and virion release). We recently established three HEV reporter systems, each covering one or two of these steps. Using these reporter systems, we identified various potential drug candidates that target different steps of the HEV life cycle. Through rigorous in vitro testing using our robust cell culture system with the genotype 3 HEV strain (JE03-1760F/P10), we confirmed the efficacy of these drugs, when used alone or in combination with existing anti-HEV drugs. This underscores their significance in the quest for an effective anti-HEV treatment. In the present review, we discuss the development of the three reporter systems, their applications in drug screening, and their potential to advance our understanding of the incompletely elucidated HEV life cycle.

## 1. Introduction

Hepatitis E virus (HEV) is a single-stranded positive-sense RNA virus with a genome spanning approximately 7.2 kilobases (kb) in length [1]. The genome structure encompasses a short 5′ untranslated region (UTR) with a 7-methylguanosine cap, three open reading frames (ORFs), and a short 3′-UTR terminated by a poly(A) tract [2,3]. Notably, ORF1 accounts for approximately two-thirds of the genome length and encodes a non-structural polyprotein that harbors multiple functional domains involved in viral replication. These domains comprise a methyltransferase (MeT), Y domain, papain-like cysteine protease (PCP), hypervariable region (HVR), X or macro domain, helicase (Hel), and RNA-dependent RNA polymerase (RdRp) [4,5]. ORF2 encodes the viral capsid protein, which manifests in the infectious, glycosylated, and cleaved forms. The glycosylated dimer form of ORF2, when secreted extracellularly, potentially serves as a decoy against humoral immunity during HEV infection [6,7]. In contrast, ORF3 encodes a small protein that is crucial for virion egress from infected cells [8,9,10], which is a functional ion channel acting as a viroporin [11]. The virus exists in two distinct particle forms: membrane-unassociated particles found in bile and feces (neHEV) and membrane-associated particles present in circulating blood and culture supernatant (quasi-enveloped HEV: eHEV) [12,13,14,15] (Figure 1A).

HEV belongs to the family *Hepeviridae*, subfamily *Orthohepevirinae*, and genus *Paslahepevirus* [1]. The members of species *Paslahepevirus balayani* have been assigned to genotype 1 HEV (HEV-1) to HEV-8 [1]. Among the five known hepatitis viruses, HEV exhibits distinctive transmission routes encompassing fecal–oral and zoonotic foodborne pathways, in addition to less common routes such as organ transplantation or the transfusion of blood products [16]. Notably, HEV is the solitary human hepatitis virus with potential for zoonotic transmission [17]. HEV-1 and HEV-2 are restricted to humans and are linked to outbreaks in developing countries where transmission occurs via the fecal–oral route, whereas HEV-3 and HEV-4 can induce zoonotic infections across a broader range of hosts and are the primary causes of sporadic and autochthonous HEV infections in developed nations [18]. HEV-3 and HEV-4 have been predominantly isolated in humans, pigs, and wild boars [1]. Variants of HEV-3 discovered in rabbits have also been isolated from humans in France [19], while HEV-3 originating from deer has affected two families in Japan [20,21]. HEV-5 and HEV-6 have exclusively been found in wild boars in Japan [22,23]. HEV-7 has been isolated in dromedary camels [24] and was associated with chronic infection in a liver transplant recipient due to the regular consumption of camel milk and meat products [25]. Conversely, HEV-8 has been identified in Bactrian camels [26,27]. Alongside certain members of the *Paslahepevirus balayani* species, the *Rocahepevirus ratti* species has also been implicated in human infections, affecting not only immunocompromised patients [28,29], but also an immunocompetent individual [30].

In addition to causing typical hepatitis, HEV infection has been linked to a wide spectrum of extrahepatic manifestations, primarily observed in immunosuppressed patients. These include neurological disorders such as Guillain–Barré syndrome, neuralgic amyotrophy, and meningoencephalitis [31]; kidney disorders such as cryoglobulinemic glomerulonephritis [32], cryoglobulinemic membranoproliferative glomerulonephritis [33], and exacerbation of IgA nephropathy [34]; and hematological disorders including thrombotic thrombocytopenic purpura [35,36,37] and aplastic anemia [38]. Other extrahepatic manifestations attributed to HEV infection comprise pancreatitis [39,40,41], myocarditis [42,43], and thyroiditis [44,45].

HEV is being recognized increasingly frequently as the primary cause of acute hepatitis. Although most HEV infections are self-limiting, immunocompromised patients can develop a chronic course [46]. The management of both chronic and acute fulminant cases relies on anti-HEV treatment [47]. Studies on treatment for chronic HEV infection have been reported concerning several drugs, such as sofosbuvir and pegylated interferon (IFN). Sofosbuvir, a nucleotide analog inhibitor of the hepatitis C virus (HCV) NS5B polymerase, exhibits inhibitory effects on HEV RNA replication both in vitro and in vivo [48]. However, its antiviral activity against HEV is comparably weaker than its efficacy against HCV replication [48]. IFN-based therapy was used as one of treatment modalities in chronic HCV infection. A previous investigation documented that various forms of IFN inhibit the replication of HEV RNA [49]. However, due to its adverse effects such as bone marrow toxicity and neuropsychiatric impacts [48,50], the use of this drug in chronic HEV infections—which mainly affect immunocompromised patients with multiple underlying diseases and morbidities—may be a major concern. Although pegylated IFN has shown some success in a limited number of patients with chronic HEV infection [51,52], it carries a risk of acute rejection in transplant recipients and subsequent graft loss [53,54]. Currently, ribavirin is the mainstay therapy [47]. The precise mechanism by which ribavirin acts against HEV remains incompletely elucidated. It was hypothesized that ribavirin may deplete guanosine triphosphate (GTP) pools, thereby leading to the inhibition of HEV replication [55]. However, the significant side effect of dose-dependent anemia caused by ribavirin use has restricted its clinical application [47]. Consequently, the need for novel, targeted anti-HEV drugs has arisen to expand the treatment options. One promising approach involves targeting distinct steps of the viral life cycle [56]: the early step (attachment and internalization), middle step (translation and RNA replication), or late step (virus particle formation and virion release) (Figure 1B).

In various virological investigations, bioluminescent reporter viruses have immense significance for drug discovery and development. They enable convenient and rapid quantification of viral replication [57,58]. Previously, we engineered an HEV replicon expressing *Gaussia* luciferase (HEV-GLuc replicon) (Figure 2A). This system facilitates the assessment of drugs inhibiting RNA replication [59,60]. To broaden the scope of drug screening while targeting the early and late steps in the HEV life cycle, we generated recombinant infectious HEV variants. These include a variant harboring the nanoKAZ gene within the ORF1 region (HEV-nanoKAZ) [61] (Figure 2B) and another incorporating a HiBiT tag in the ORF2 region (HEV-HiBiT) [62] (Figure 2C). Both models closely emulate physiological viral infection conditions, which is a rarity in HEV research. The application of these three reporter systems led to the identification of several candidate drugs that target various steps in the HEV life cycle. Their efficacy has been verified in vitro through the utilization of our robust cell culture system, wherein the virus was adapted for enhanced replication efficiency in PLC/PRF/5 cells (specifically the HEV-3 strain JE03-1760F/P10, generated following 10 consecutive passages of the wild-type strain) [63,64], both in individual treatments and in combination with existing anti-HEV drugs [61,62,65], underscoring their utility in identifying potential anti-HEV agents.

This review provides an overview of the recent developments in our HEV reporter systems, their utility as drug screening platforms, and their potential as valuable tools for studying the HEV life cycle.

## 2. The HEV-GLuc Replicon: An Advanced Tool for Monitoring HEV Replication and Drug Screening

The use of replication reporter systems incorporating the HEV-GLuc replicon offers a distinct advantage in monitoring the replication of the targeted replicon virus. This system provides a straightforward and facile method to screen for potential antiviral drugs. GLuc, a 19.9-kDa secretory luciferase, was originally extracted from *Gaussia princeps*, a marine copepod [66]. It was proven to be effective as a reporter protein for tracking the replication of several HEV strains [67,68,69,70]. The primary HEV replicon housing GLuc, which is widely employed and initially reported, belongs to genotype 3 (Kernow-C1 p6/GLuc). In this configuration, the GLuc sequence replaces a portion of the sequence downstream of the ORF2 start codon [71] (Table 1).

Parallels were effectively employed for constructing HEV replicons in HEV-1 [67,72], rat HEV [73], HEV-4 [72], swine HEV-3 [72,74], and rabbit HEV-3ra [75] (Table 1). In addition, a variant of Kernow-C1 p6/GLuc was engineered by introducing hemagglutinin or V5 tags into ORF1 (Table 1) [76]. Among these constructs, Kernow-C1 p6/GLuc has predominantly been employed for drug screening, leading to the identification of anti-HEV activity drugs, such as deptropine [77], gemcitabine [78], and isocotoin [79] (Table 1). Various HEV replicons harboring *Gaussia* luciferase have been used in diverse HEV studies. These include analyzing the role of inserting the human S17 ribosomal protein sequence for growth advantage [71], assessing the significance of the C-terminal 52 amino acids [74], identifying potential determinants of the host range [67], exploring viral regulatory elements and intracellular genome dynamics [72], and identifying candidate viral factories [76].

**Table 1 viruses-15-01989-t001:** Various HEV replicons harboring *Gaussia* luciferase.

Strain	Genotype (Origin)	Application for:		Reference
		Drug Screening	Analysis of Life Cycle	
Kernow-C1 p6	HEV-3 (human)	NA	Role of insertion of human S17 ribosomal protein sequence in growth advantage	[71]
		Deptropine	NA	[77]
		Gemcitabine	NA	[78]
		Isocotoin	NA	[79]
HEV83-2-27	HEV-3 (swine)	NA	Importance of C-terminal 52 amino acids for HEV life cycle	[74]
Sar55/S17	HEV-1 (human)	NA	Identification of possible determinants of host range	[67]
LA-B350	HEV-C1 (rat)	NA	Establishment of subgenomic replicon for various HEV studies	[73]
Sar55	HEV-1 (human)	NA	Identification of viral regulatory elements and intracellular genome dynamics	[72]
SHEV3	HEV-3 (swine)	NA	Identification of viral regulatory elements and intracellular genome dynamics	[72]
TW6196-E	HEV-4 (human)	NA	Identification of viral regulatory elements and intracellular genome dynamics	[72]
JE03-1760F/P10	HEV-3 (human)	Ciprofloxacine	NA	[60]
Kernow-C1 p6 (with HA- or V5-tagged ORF1)	HEV-3 (human)	NA	Identification of candidate HEV factories	[76]
HEV-3ra LR	HEV-3ra (rabbit)	NA	Investigation of the impact of ribavirin-treatment-failure-associated RdRp mutations of human HEV-3 on in vitro replication of HEV-3ra	[75]

NA, not available; HEV, hepatitis E virus; HEV-3, genotype 3 HEV; HEV-1, genotype 1 HEV; HEV-4, genotype 4 HEV; HA, hemagglutinin; V5, a tag protein; HEV-3ra, a subtype of HEV sequences derived from rabbits (and closely related sequences from humans) within genotype 3; RdRp, RNA-dependent RNA polymerase.

In the previous study, we engineered an HEV replicon incorporating GLuc based on the infectious cDNA clone pJE03-1760F/P10, an HEV-3 strain adapted to cell culture conditions. This strain, generated after 10 consecutive passages of the wild-type strain, exhibited heightened HEV production compared to the wild-type infectious cDNA clone (pJE03-1760F/wt) [63]. This particular strain is well suited for reporter assays because of its robust virus production. The construction of pJE03-1760F/P10-GLuc involved the disruption and replacement of the ORF3 and ORF2 genes with GLuc. Consequently, this construct hinders the expression of ORF2 and the multifunctional ORF2-overlapping protein. ORF3 [60].

Agents known for their anti-HEV activity, such as ribavirin [55], mycophenolic acid [80], sofosbuvir [81], and IFN-α2b [52], can effectively suppress GLuc expression in pJE03-1760F/P10-GLuc RNA-transfected PLC/PRF/5 cells. This outcome underscores the effectiveness of the HEV replication reporter system for evaluating anti-HEV drug activity. Screening a library of 767 Food and Drug Administration (FDA)-approved drugs using the HEV-GLuc replicon system revealed several compounds with varying degrees of inhibition against HEV activity. Notably, ciprofloxacin (Table 1) exhibited sufficient inhibitory activity with minimal toxicity. Although not as potent as ribavirin, ciprofloxacin effectively curbed HEV growth in cultured cells [60].

According to various reports on chronic HEV infections, ribavirin therapy achieved a sustained virological response (SVR), as evidenced by undetectable serum HEV RNA levels for at least six months post-treatment cessation, in approximately 80% of patients treated with monotherapy [46,82,83]. The exploration of drug combinations to enhance SVR rates is promising. To this end, the HEV-GLuc replicon facilitated the screening of antiviral compound classes, such as IFNs (IFN-α2b and IFN-λ1-3), 2-methyl ribosides, and 4-azido ribosides, revealing some members with activity against HEV [60,84,85]. Notably, the combination of 2′-C-methylguanosine and ribavirin demonstrated a synergistic effect in inhibiting HEV replication [59].

It is important to exercise caution when interpreting the results from HEV replication reporter systems using GLuc. While these systems are powerful tools for identifying potential anti-HEV candidates, the observed inhibitory effects may not necessarily translate to actual virus growth inhibition. Therefore, verification using in vitro viral growth assays is essential. We evaluated the identified drugs in our robust cell culture system using PLC/PRF/5 cells, which supported a more efficient propagation of HEV than other hepatoma cells [64] over a long period. Over a substantial time frame, the combination of 2′-C-methylguanosine, 2′-C-methylcytidine, and sofosbuvir, along with ribavirin, demonstrated additive effects in inhibiting HEV growth and eliminating HEV from cultured cells. Similarly, the combination of 2′-C-methylguanosine and sofosbuvir, paired with four IFNs, displayed additive effects in inhibiting HEV growth and eradicating HEV genomes in cultured cells [59]. These findings suggest that the phosphoramidate prodrug of both 2′-C-methyluridine and 2′-C-methylguanosine monophosphates, possessing a 2′-hydroxy group, holds promise as a potential anti-HEV drug, either alone or in combination with ribavirin and/or IFNs.

In conclusion, the HEV-GLuc replicon serves not only as a robust platform for drug screening, but also as an effective reporter for monitoring HEV RNA replication. This study offers valuable insights into potential anti-HEV candidates and their mechanisms of action, emphasizing the need for further validation using in vitro viral growth assays.

## 3. The Recombinant Infectious HEV-nanoKAZ

Currently, the predominant method for screening candidate novel anti-HEV drugs worldwide involves the utilization of subgenomic replicons. These replicons, including one established in our laboratory [60] (depicted in Figure 2, upper panel), serve as the sole available approach for exploring potential drug candidates targeting HEV. These replicons facilitate the examination of viral RNA replication, while avoiding the production of infectious particles. As a result, they are suitable for assessing drug efficacy against HEV RNA replication [60].

In an effort to expand the scope of drug screening to search for candidate drugs targeting the early steps of the HEV life cycle, we developed a recombinant infectious HEV harboring a nanoKAZ gene inserted in the HVR of the ORF1 region. This construct, referred to as HEV-nanoKAZ, closely mimics the physiological conditions of viral infections [61]. To create HEV-nanoKAZ, we employed the infectious cDNA clone of HEV-3, pJE03-1760F/P10, as a template, which is known for its robust HEV production, similar to our HEV-GLuc replicon. The nanoKAZ gene, a 171-amino-acid-mutated 19 kDa component of *Oplophorus* luciferase, was inserted into the HVR. This gene, a recent addition to luciferase enzyme systems [86,87,88], features an amino acid sequence that is identical to NanoLuc yet with a distinct nucleotide sequence [86,89]. HVR has been recognized to accommodate naturally acquired insertions in both acute [90,91] and chronic HEV patients [71,92,93,94] as well as in swine [95]. In addition, genetic recombination leads to insertions [76,96].

Previous studies by Szkolnicka et al. [96] and Metzger et al. [76] have demonstrated that epitope tags inserted into the HVR did not hinder viral replication. In a report by Szkolnicka et al. [96], the site was determined via transposon-mediated random insertion coupled with selection in a subgenomic replicon system using the hemagglutinin or NanoLuc tag to identify candidates for HEV replication complexes and for the subcellular localization and analysis of the polyprotein processing of ORF1 (Table 2). In our research, the nanoKAZ insertion was positioned at five distinct sites within the HVR, which are sites that were previously identified in HEV-3-infected humans or pigs with insertions of at least 30 amino acids [71,90,91,92,93,94,95] (Figure 3; Table 2). Among these five engineered HEV constructs, one with insertion site number 3 (Figure 3) demonstrated the ability to replicate and maintain stable nanoKAZ insertion across consecutive passages [61].

Characterization of HEV-nanoKAZ revealed that both forms of viral particles, eHEV-nanoKAZ and neHEV-nanoKAZ, were infectious [61]. The infectivity was confirmed through inoculation of eHEV-nanoKAZ and neHEV-nanoKAZ to PLC/PRF/5 cells, where HEV ORF2 protein expression was detectable via immunofluorescence assay following the inoculation, and the luciferase activity gradually increased over time in the lysates of the inoculated cells [61]. This dual infectivity offers a functional tool for various HEV studies requiring both viral particle forms, including investigations into unknown HEV receptors and elucidation of host factors crucial for HEV entry. In addition to its efficient propagation in PLC/PRF/5 cells, eHEV-nanoKAZ also exhibited replication across a wide range of cell types, spanning cancer cell lines (HepG2/C3A, A549, and Caco-2) to normal hepatocytes (PXB-cells) [61], where nanoKAZ is specifically produced. This versatility supports its potential application in diverse cell culture conditions for future HEV research [61].

We evaluated the sensitivity of eHEV-nanoKAZ to the anti-HEV agents, sucrose and ribavirin. Sucrose, known to inhibit clathrin-dependent endocytosis, is an HEV entry inhibitor, given that HEV entry depends on clathrin-mediated endocytosis rather than caveola-mediated endocytosis [14]. Conversely, ribavirin, which is currently employed in treating certain HEV infections such as acute fulminant or chronic cases [47], is an inhibitor of HEV RNA replication. Treatment with sucrose or ribavirin in eHEV-nanoKAZ-inoculated PLC/PRF/5 cells led to reductions in intracellular luciferase activity in a dose-dependent manner. Intriguingly, neither genistein (a caveola-mediated endocytosis inhibitor [97], used as a negative control for sucrose) nor lomibuvir (an inhibitor of RNA polymerase NS5B of HCV), which have been used as negative controls for ribavirin in previous studies [59,60], affected the intracellular luciferase activity. This suggests that the reporter system covers the inhibition of HEV entry and HEV RNA replication. To validate the utility of eHEV-nanoKAZ for drug screening, we screened a commercially available FDA-approved drug library. The inhibitory effect of the drugs that passed the screenings on intracellular luciferase activity was comparable to that of sucrose and ribavirin, which served as reference drugs. This confirmed the efficacy of the drug-screening platform [61].

During the screening process, four drugs exhibited activity: gefitinib, chlorpromazine, azithromycin, and ritonavir. The inhibition of intracellular luciferase activity displayed a dose-dependent pattern. The combination of the eHEV-nanoKAZ and HEV-GLuc replicon systems indicated that these four drugs did not affect HEV RNA replication. Thus, they likely interfere with the early steps in the HEV life cycle, such as attachment to cell receptors, internalization, or uncoating [61]. Chlorpromazine inhibited HEV entry through clathrin-mediated endocytosis [14], whereas azithromycin was recently identified as a potent inhibitor of HEV replication and viral protein expression in cell culture [98]. This reaffirms the reliability of the screening system in identifying potential anti-HEV drugs.

Further investigation confirmed that ritonavir, the most potent of the four hit drugs, inhibited HEV internalization. Consequently, we examined its combination with ribavirin, considering their targeting of different steps in the HEV life cycle, as a potential novel strategy against chronic HEV infection. Notably, this combination exhibited more robust inhibitory activity than ribavirin monotherapy, both in the eHEV-nanoKAZ reporter assay and in long-term cell culture assessments. This efficacy was consistent for both HEV-3 and HEV-4 [65], reinforcing the value of the HEV-nanoKAZ system in identifying potential treatment options to combat HEV infections.

In summary, the HEV-nanoKAZ system has been effectively applied for drug screening, validating anti-HEV drugs, and evaluating the effectiveness of drug combinations in inhibiting HEV growth in cell cultures (Table 2).

## 4. The Recombinant Infectious HEV-HiBiT

To enhance our drug screening system, we generated an additional reporter HEV (HEV-HiBiT) by introducing a truncated HiBiT tag derived from NanoLuc luciferase to the 3′ end of the ORF2 coding sequence [62]. This modification aimed to identify potential drug candidates that inhibit late steps in the HEV life cycle, specifically particle formation or virion release. For the construction of this reporter virus, we also used the infectious cDNA clone of pJE03-1760F/P10, a cell-culture-adapted strain with robust HEV production capabilities, as the template. HiBiT is a compact 11-amino-acid split-reporter tag derived from NanoLuc binary technology (NanoBiT) that exhibits a strong affinity for the split-LgBiT (158 amino acids) reporter [99]. This HiBiT-tagged system enables the facile detection and quantification of HiBiT-tagged proteins using the Nano-Glo assay system [100].

We introduced a tag within the capsid to monitor HEV particle formation and release. Specifically, we integrated the HiBiT sequence, which included two consecutive glycine–serine linker sequences and two stop codons, into the 3′ end of the ORF2 coding sequence (Figure 2C; Table 3). This insertion site choice was grounded in our earlier work that demonstrated a functional HEV genome with a C-terminal FLAG tag on the ORF2 protein [101]. The 3′-terminal region of the ORF2 sequence plays a pivotal role in replication [102,103], and the C-terminus of the ORF2 protein is critical for viral genome encapsidation and particle stabilization [74]. Furthermore, cis-reactive elements (CREs), which are composed of stem loops within secondary RNA structures found in non-coding regions, are indispensable for viral replication [104]. In a previous study, to develop an HEV-like particle featuring a FLAG-tagged ORF2 protein, we integrated a repeat of the 3′-terminal ORF2 sequence (nt 7092–7151 (60 nt)) downstream of the FLAG sequence in the genome of JE03-1760F/P10 [99] (Table 3). This construct elucidated the maturation of enveloped and non-enveloped HEVs, demonstrating that ORF2 proteins associated with enveloped HEV have an intact C-terminus, whereas non-enveloped HEV-associated ORF2 proteins are C-terminally truncated [101] (Table 3). The insertion of the FLAG-tag sequence disrupted the CRE-containing secondary structure, leading to the inability of the FLAG-tagged HEV-like particles to propagate in cell culture [101]. The sequence repeat preserves the crucial cis-acting replication element located at the 3′ end of the ORF2-coding region. As illustrated in Figure 4, the anticipated secondary structures at the C-terminus of ORF2 and at the 3′ UTR will remain intact in the recombinant HEV-HiBiT construct with the 60 nt 3′-terminal of the ORF2 repeat (lower panel), while these secondary structures are compromised when no 60 nt 3′ terminal of the ORF2 repeat is inserted (middle panel). Therefore, in the present study, we introduced a repeat of the 60 nt 3′-terminal ORF2 coding sequence (nt 7092–7151) between the two stop codons following the HiBiT sequence and the authentic stop codon of the ORF2 coding sequence to uphold the essential stem loop structure required for HEV replication [62].

The selection of the linker was influenced by prior findings indicating the importance of a peptide linker connecting the analytical target and tag [104], which aligns with the outcomes of our preliminary investigation. In brief, we transfected RNA transcripts of pHEV3b-HiBiT (no linker), pHEV3b-HiBiT (1 × linker), pHEV3b-HiBiT (2 × linker), and pHEV3b_GAA (a replication-defective mutant) into PLC/PRF/5 cells to assess their replication efficiency, along with RNA transcripts of pHEV3b for reference. HEV growth kinetics were then observed for 28 days. The HEV RNA titer in the culture supernatants of cells transfected with the linker-less construct exhibited faster growth than constructs with one or two linkers (Figure 5A). Conversely, luciferase activity in the culture supernatants of cells transfected with the linker-less construct at 28 days post-transfection (dpt) was lower than that in the cells transfected with constructs with one or two linkers (Figure 5B). Furthermore, gel electrophoresis revealed that the length of the reverse transcription-polymerase chain reaction (RT-PCR) products from purified RNA molecules in culture supernatants at 28 dpt was longer for the constructs with one or two linkers than for the linker-less construct, which maintained the same length as the wild-type virus. This indicated that the linker-less construct lost the HiBiT insertion, whereas the constructs with one or two linkers retained the HiBiT insertion until at least 28 dpt (Figure 5C), which was corroborated by the sequence analysis results. In addition, the insertion was lost as early as 8 dpt for the linker-less construct. These findings collectively suggest that the insertion of the HiBiT sequence into the virus genome without linker sequences renders it less stable, leading to its rapid elimination post-transfection, whereas the insertion of the HiBiT sequence along with one or two linker sequences remains sustainable. Considering that the luciferase activity from the construct with two linkers surpassed that of the construct with one linker, we opted for the construct with two linkers for our HEV-HiBiT study. Furthermore, the construct incorporating two linkers sustained stable HiBiT insertion through three consecutive passages while maintaining HiBiT expression levels [62]. This strategy facilitated the successful creation of an infectious virus bearing a HiBiT tag.

HEV produces at least two distinct protein forms translated from ORF2: ORF2s, the secreted form, initiated from the first AUG codon (Met1) with its 23 N-terminal amino acids cleaved via signal peptidase, and ORF2c, the capsid protein, initiated from an internal AUG (Met16) [6,7]. The HiBiT tag is fused to ORF2s (ORF2s-HiBiT) and positioned externally, whereas the HiBiT tag fused to ORF2c (ORF2c-HiBiT) resides within the lipid membrane of virus particles (eHEV-HiBiT) and is exposed on the surface of the HEV capsid [62].

For drug screening, ORF2s-HiBiT was separated via sucrose density gradient centrifugation. Recognizing this substantial workload, we designed a variant with dual mutations, replacing Met with Val at the ORF2s initiation codons of both ORF2s and ORF2c (HEV-HiBiT/ΔORF2s) [62]. This variant effectively prevented the expression of secretory ORF2 proteins while preserving HEV-HiBiT production. Despite the overlap of ORF2 and ORF3, these mutations did not alter the ORF3 amino acid sequence. This variant displays sensitivity to not only a drug inhibiting HEV release (GW4869) and an accelerator of HEV release (bafilomycin A1), but also HEV RNA replication (ribavirin), rendering it a valuable drug screening platform [62].

Furthermore, integrating this system with our HEV-GLuc replicon system facilitated the distinction between HEV release and HEV RNA replication. The formation and release of eHEV closely involve the endosomal sorting complexes required for transport (ESCRT) machinery and exosomal pathway [15,105,106]. The depletion of ESCRT and exosomal pathway constituents, such as tumor susceptibility gene 101 (Tsg101), Ras-associated binding 27A (Rab27A), or hepatocyte growth factor-regulated tyrosine kinase substrate (Hrs), using small interfering RNA (siRNA) led to reduced luciferase activity in the culture supernatants of siRNA-treated PLC/PRF/5 cells transfected with pHEV-HiBiT/ΔORF2s RNA [62]. These findings emphasize its potential utility for examining the HEV life cycle, particularly for analyzing virus release mechanisms.

## 5. Conclusions

In the development of novel antiviral drugs, a strategic approach involves targeting distinct steps within the viral life cycle: the early step, encompassing attachment and internalization; the middle step, involving translation and RNA replication; and the late step, centered around virus particle formation and subsequent virion release. To this end, our laboratory has recently established three distinct HEV reporter systems, the HEV-nanoKAZ, HEV-GLuc replicon, and HEV-HiBiT systems, each offering varying degrees of coverage for drug screening. The robust and consistent outcomes obtained through the implementation of these three HEV reporter systems further accentuate their utility as streamlined platforms for drug screening.

The amalgamation of these engineered viruses presents an opportunity for comprehensive screening, facilitating the identification of potential anti-HEV drugs/compounds that effectively act against specific stages of HEV infection. Simultaneously, this strategy empowers us to meticulously dissect the intricate mechanisms through which the identified therapeutic agents exert their inhibitory effects. The near-physiological conditions of viral infection accurately replicated via HEV reporter systems confer a distinct advantage in drug screening. Consequently, these systems assume a pivotal role in not only drug development pursuits, but also in the investigation of the nuances of the life cycle of HEV. For instance, they enable the exploration of enigmatic receptors, intricate virus–host interactions, and pivotal host factors that underpin proficient HEV replication.

The judicious utilization of these innovative tools holds the promise of enriching our understanding of HEV biology and paving the way for the advancement of efficacious therapeutic interventions against HEV infections.

## Figures and Tables

**Figure 1 viruses-15-01989-f001:**
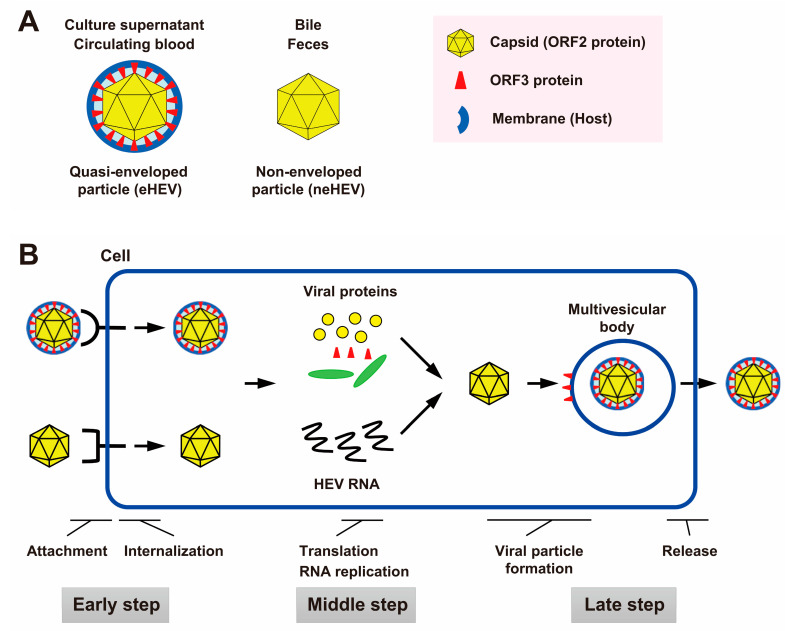
(**A**) Schematic representation of two distinct forms of HEV particles. The membrane-unassociated form present in bile and feces is referred to as neHEV, while the membrane-associated form found in circulating blood and culture supernatant, covered by a cellular membrane and the open reading frame 3 (ORF3) protein, is termed quasi-enveloped HEV (eHEV). (**B**) The HEV life cycle and potential drug targets. Novel antiviral drugs can be designed to target specific steps in the viral life cycle, including the early step (attachment and internalization), middle step (translation and RNA replication), and late step (virus particle formation and virion release). These systems mimic physiological viral infection conditions, making them advantageous for drug screening and enhancing our understanding of the HEV life cycle.

**Figure 2 viruses-15-01989-f002:**
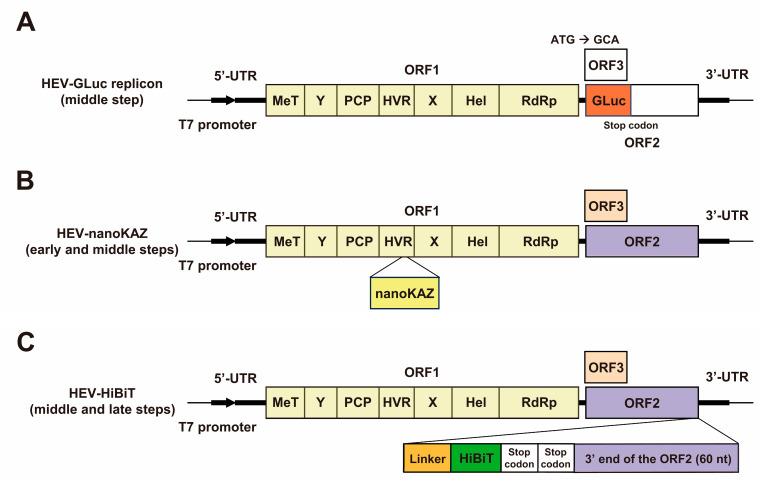
Schematic representation of HEV reporter systems developed from an infectious cDNA clone (pJE03-1760F/P10) of genotype 3 HEV (HEV-3), a strain that underwent 10 consecutive passages of the wild-type strain and has enhanced HEV production in comparison to the wild-type infectious cDNA clone [63]. (**A**) HEV replicon expressing *Gaussia* luciferase (HEV-GLuc) [60]. To construct the pJE03-1760F/P10-GLuc, the ORF3 and ORF2 genes were disrupted and replaced with GLuc. (**B**) Infectious recombinant HEV harboring nanoKAZ gene within the HVR of ORF1 (HEV-nanoKAZ) [61]. (**C**) Infectious recombinant HEV with a HiBiT tag placed at the 3′ end of the ORF2 coding sequence (HEV-HiBiT) [62]. The HiBiT tag sequence contains two tandem glycine–serine linker sequences and two stop codons. The 3′-end 60 nt sequence of ORF2 was inserted after the HiBiT tag sequence. MeT, methyl transferase; Y, Y domain; PCP, papain-like cysteine protease; HVR, hypervariable region; X, X or macro domain; Hel, helicase; RdRp, RNA-dependent RNA polymerase.

**Figure 3 viruses-15-01989-f003:**
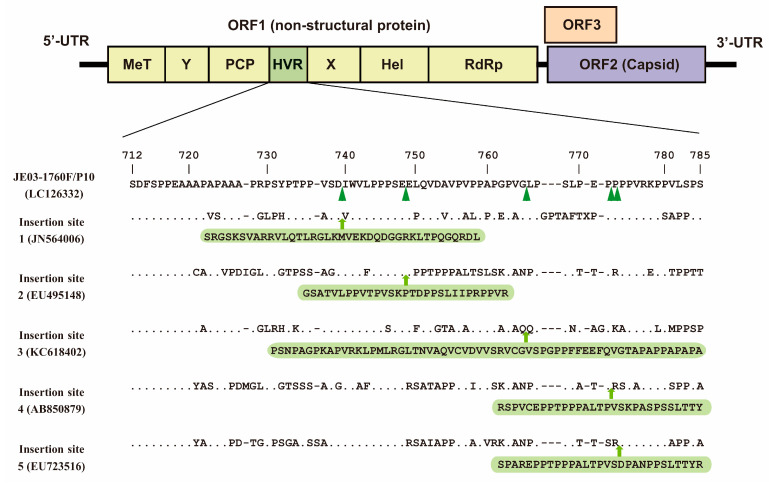
Depiction of representative candidate insertion sites of the nanoKAZ gene in the HVR. The insertion was placed at 5 distinct sites within HVR that have been reported in various HEV-3-infected humans or pigs with at least 30 amino acid insertions. Among the five recombinant HEV constructs, the construct harboring the nanoKAZ gene at insertion site number 3 was chosen due to it having the most efficient replication and stable insertion during consecutive passages.

**Figure 4 viruses-15-01989-f004:**
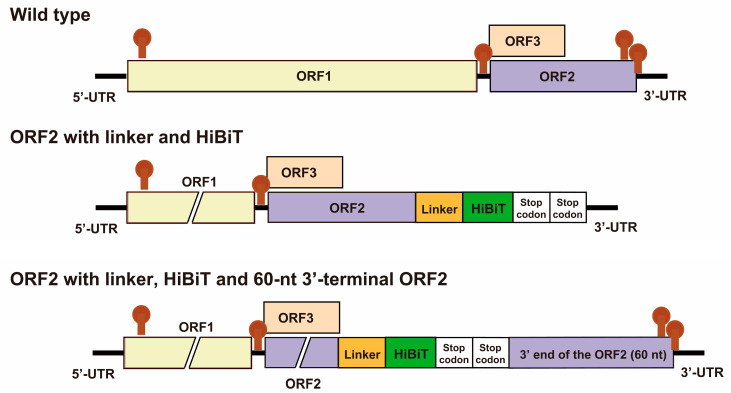
Schematic representation of 4 predicted secondary (stem-loop) structures in the genome of wild-type JE03-1760F/P10 (**top** panel) in comparison to the recombinant HEV-HiBiT construct either lacking (**middle** panel) or containing (**bottom** panel) the 60 nt 3′ terminal of ORF2 repeat. The figure emphasizes that essential secondary structures at the C-terminus of ORF2 and at the 3′ UTR, crucial for HEV replication, remain intact in the recombinant HEV-HiBiT construct with the 60 nt 3′ terminal of ORF2-repeat (**lower** panel), while these two structures are disrupted if the 60 nt segment is absent (**middle** panel).

**Figure 5 viruses-15-01989-f005:**
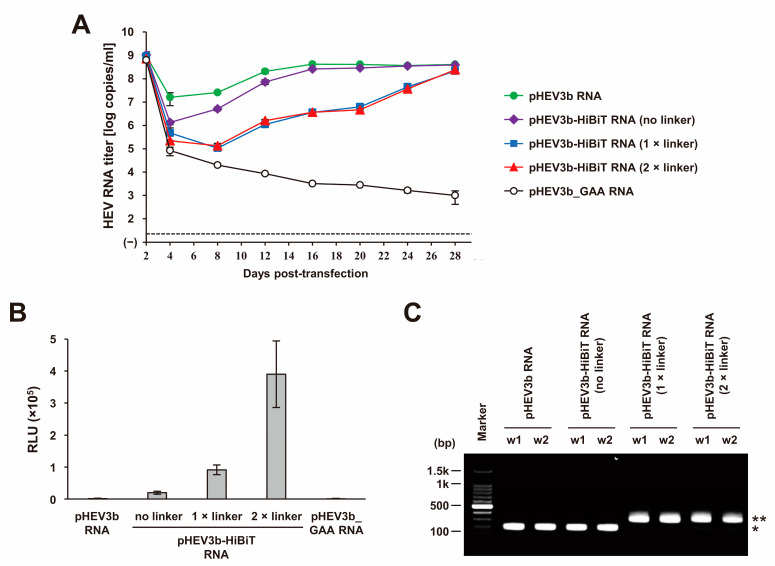
A comparative analysis of HEV-HiBiT constructs with and without linker sequences. (**A**) Quantification of HEV RNA in culture supernatants. Different RNA transcripts, including those of pHEV3b-HiBiT (no linker), pHEV3b-HiBiT (1 × linker), pHEV3b-HiBiT (2 × linker), and pHEV3b_GAA (a replication-defective mutant), were transfected to PLC/PRF/5 cells to assess replication efficiency, along with RNA transcripts of pHEV3b as a control. The HEV growth kinetics were observed for 28 days. The dotted horizontal line represents the limit of detection determined via real-time RT-PCR assay used in this study at 2 × 10^1^ RNA copies/mL. (**B**) Luciferase activity in culture supernatants of PLC/PRF/5 cells transfected with the RNA transcripts of pHEV3b, pHEV3b-HiBiT (no linker), pHEV3b-HiBiT (1 × linker), pHEV3b-HiBiT (2 × linker), and pHEV3b_GAA at 28 days post-transfection (dpt). Culture supernatants were subjected to sucrose density gradient centrifugation. Fraction 13 was treated with 0.1% digitonin and used for the measurement of the luciferase activity. The data are presented as the mean ± standard deviation for two wells each. (**C**) Gel electrophoresis of the RT-PCR products, covering the 3′-terminal ORF2 sequence containing the linker sequence, HiBiT insertion, two stop codons, and repeated 3′ end 60 nt ORF2 coding sequence from purified RNA molecules in culture supernatants at 28 dpt. The size of PCR product without HiBiT insertion is 199 base pairs (bp) (*), while the size of the PCR product with intact HiBiT insertion is 322 bp (**).

**Table 2 viruses-15-01989-t002:** The comparison of reported infectious HEV cDNA clones harboring a reporter in the ORF1.

Strain (Genotype)	Insertion Site	Strategy to Select Insertion Site	Tag	Application	Reference
HEV83-2-27 (HEV-3)	HVR	Transposon-mediated random insertion coupled with selection in a subgenomic replicon system	Hemagglutinin or NanoLuc	Identification of candidate HEV replication complexes Subcellular localization and analysis of polyprotein processing of ORF1	[96]
JE03-1760F/P10 (HEV-3)	HVR	Based on insertion sites that have been reported in HEV-3-infected humans or pigs with ≥30-aa insertion	nanoKAZ	Drug screening to search for candidate anti-HEV drug Validation of anti-HEV activity of identified drug Evaluation of effectiveness of drug combination	[61]

HEV, hepatitis E virus; ORF1, open reading frame 1; HEV-3, genotype 3 HEV; HVR, hypervariable region.

**Table 3 viruses-15-01989-t003:** The comparison of reported infectious HEV clones harboring a reporter in the ORF2.

Strain (Genotype)	Tag	Strategy	Application	Findings	Reference
JE03-1760F/P10 (HEV-3)	FLAG	Two tandem glycine–serine linker sequences and FLAG tag were inserted at the 3′ end of the ORF2 sequence, followed by two stop codons and the 3′-terminal 60 nt ORF2 sequence (nt 7091–7151).	Study on the maturation of the enveloped and non-enveloped HEVs.	ORF2 proteins associated with enveloped HEV have an intact C-terminus, but those associated with non-enveloped HEV are C-terminally truncated.	[101]
JE03-1760F/P10 (HEV-3)	HiBiT	Two tandem glycine–serine linker sequences and HiBiT tag were inserted at the 3′ end of the ORF2 sequence, followed by two stop codons and the 3′-terminal 60 nt ORF2 sequence (nt 7091–7151).	Confirmation of applicability for drug screening. Analysis of HEV release using an ORF2s-defective variant.	The reporter HEV replicated efficiently in PLC/PRF/5 cells, produced membrane-associated particles, and was genetically stable and infectious.	[62]

HEV, hepatitis E virus; ORF2, open reading frame 2; HEV-3, genotype 3 HEV; nt, nucleotide.

## Data Availability

All data are presented in the manuscript.

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
