# Peer review of "Three Distinct Reporter Systems of Hepatitis E Virus and Their Utility as Drug Screening Platforms"

_viruses, 2023, doi:10.3390/v15101989_

Round 1
Reviewer 1 Report
These authors have developed reporter systems for hepatitis C virus which allows them to determine the effect of drugs on early steps, middle steps, and late steps in virus replication. This system allows them to screen drugs which have potential utility in hepatitis C infections. This work involves a relatively complicated viral molecular biology. The authors have explained their approach in significant detail. This reviewer has no recommendations or suggestions which would improve the quality of this and manuscript. I assume the details are essential for interested investigators using this technology to make use of this information. In addition, I would assume that drug companies with large numbers of untested drugs would be interested in this technology.
Author Response
These authors have developed reporter systems for hepatitis C virus which allows them to determine the effect of drugs on early steps, middle steps, and late steps in virus replication. This system allows them to screen drugs which have potential utility in hepatitis C infections. This work involves a relatively complicated viral molecular biology. The authors have explained their approach in significant detail. This reviewer has no recommendations or suggestions which would improve the quality of this and manuscript. I assume the details are essential for interested investigators using this technology to make use of this information. In addition, I would assume that drug companies with large numbers of untested drugs would be interested in this technology.
Response: Thank you for your favorable evaluation to our manuscript.

Reviewer 2 Report
HEV is one of the leading causes of acute viral hepatitis. The World Health Organization estimates that around 20 million people are infected annually, and the overall mortality rate ranges from 0.2% to 4%. Knowing the life cycle of the HEV and developing an effective treatment is essential for human health. Premadharsini et all. established three HEV reporter systems to identify potential drug candidates that target different steps of the HEV life cycle. They confirmed the efficacy of these drugs when used alone or in combination with existing anti-HEV drugs. This underscores their significance in the quest for an effective anti-HEV treatment. In 22 the present review, we discuss the development of the three reporter systems, their application in 23 drug screening, and
L 20 – It is not clear which are these robust cell culture systems. Please, rewrite the sentence.
L89-105 Please provide more information on the mechanisms at the molecular level regarding the various anti-HEV drugs
L118 –120 Please, give more details about the nanoKAZ gene and HiBiT tag! The steps of the investigation are not clear. We need more details about the reported systems and the cell culture system.
L163 In the present study, we engineered an HEV replicon incorporating GLuc based. This is a review paper and you discussed your previous research, please rewrite the sentence.
Can you make a scheme for HEV replicon incorporating GLuc based on the infectious cDNA clone pJE03-1760F/P10 and replication in a cell culture system?
L249- 250 I am concerned about how the information is presented in your review. You present the information as if it were a research paper, but without showing concrete evidence. I think this significantly complicates the perception of the data you present.
Author Response
HEV is one of the leading causes of acute viral hepatitis. The World Health Organization estimates that around 20 million people are infected annually, and the overall mortality rate ranges from 0.2% to 4%. Knowing the life cycle of the HEV and developing an effective treatment is essential for human health. Premadharsini et all. established three HEV reporter systems to identify potential drug candidates that target different steps of the HEV life cycle. They confirmed the efficacy of these drugs when used alone or in combination with existing anti-HEV drugs. This underscores their significance in the quest for an effective anti-HEV treatment.
Response: We appreciate your favorable assessment and valuable comments to improve our manuscript. Our point-to-point responses to your specific comments are presented below:
L 20 – It is not clear which are these robust cell culture systems. Please, rewrite the sentence.
Response: In accordance with your comment, the phrase “with the HEV-3 strain (JE03-1760F/P10)” was incorporated (line 21). In addition, we added the following sentences with the relevant reference [63, 64) to describe our robust cell culture system in the Introduction section (lines 168–170) “our robust cell culture system, wherein the virus was adapted for enhanced replication efficiency in PLC/PRF/5 cells (specifically the HEV-3 strain JE03-1760F/P10, generated following 10 consecutive passages of the wild-type strain) [63, 64], ...”
L89-105 Please provide more information on the mechanisms at the molecular level regarding the various anti-HEV drugs.
Response: Descriptions regarding the mechanisms at the molecular level concerning various anti-HEV drugs have been added, including inhibition mechanisms for sofosbuvir (lines 92–93), interferon (lines 96–97), and ribavirin (lines 104–106).
L118 –120 Please, give more details about the nanoKAZ gene and HiBiT tag! The steps of the investigation are not clear. We need more details about the reported systems and the cell culture system.
Response: The details about the nanoKAZ gene have been described in lines 433–436 (Section 3: The Recombinant Infectious HEV-nanoKAZ), and those about the HiBiT tag have been described in lines 564–566 (Section 4: The Recombinant Infectious HEV-HiBiT). To avoid repetition, we would like to keep the description as it is.
Regarding the steps of the investigation using our three HEV reporter systems, HEV-GLuc replicon system facilitates the assessment of drugs inhibiting RNA replication (lines 115–116 and 399; Fig. 2A), HEV-nanoKAZ system facilitates the assessment of drugs inhibiting early and middle steps—entry and RNA replication—(lines 117 and 525–526; Fig. 2B), and HEV-HiBiT system facilitates the assessment of drugs inhibiting middle and late steps—RNA replication, and particle formation and virus release) (lines 117, 561–562, and 683–685; Fig. 2C).
Regarding the cell culture system, it has been described in the revised version of this manuscript lines 168-170: “our robust cell culture system, wherein the virus was adapted for enhanced replication efficiency in PLC/PRF/5 cells (specifically the HEV-3 strain JE03-1760F/P10, generated following 10 consecutive passages of the wild-type strain) [63, 64], …”
L163 In the present study, we engineered an HEV replicon incorporating GLuc based. This is a review paper and you discussed your previous research, please rewrite the sentence.
Response: We revised the sentence accordingly: “In the previous study, we engineered an HEV replicon incorporating GLuc based …” (line 231).
Can you make a scheme for HEV replicon incorporating GLuc based on the infectious cDNA clone pJE03-1760F/P10 and replication in a cell culture system?
Response: The ORF2 region is likely more restrictive to heterologous sequence insertion given the tight and conserved protein-protein interactions required for capsid assembly. The large size of GLuc insertion makes it difficult to be inserted into ORF2 region. In addition, formation of infectious particle is difficult, therefore, we chose to construct the infectious cDNA clone with much smaller tag (HiBiT) inserted into the ORF2 region, as described in detail in reference no. 62 (Nagashima et al., J Virol 2023).
L249- 250 I am concerned about how the information is presented in your review. You present the information as if it were a research paper, but without showing concrete evidence. I think this significantly complicates the perception of the data you present.
Response: To avoid confusion, we revised the sentence accordingly (lines 503–508): “Characterization of HEV-nanoKAZ revealed that both forms of viral particles, eHEV-nanoKAZ and neHEV-nanoKAZ, were infectious [61]. The infectivity was confirmed through inoculation of eHEV-nanoKAZ and neHEV-nanoKAZ to PLC/PRF/5 cells where HEV ORF2 protein expression was detectable by immunofluorescence assay following the inoculation, and the luciferase activity gradually increased over time in the lysates of the inoculated cells [61]”.

Reviewer 3 Report
It is an interesting review discussing the reporter culture systems for HEV. The authors discussed the HEV-GLuc Replicon, the recombinant infectious HEV-nanoKAZ, and the recombinant Infectious HEV-HiBiT.
Some points to improve the quality of review
1) Reporter culture system for studying extrahepatic HEV manifestation.
2) Reporter HEV that used for in vivo animal model.
Moderate language editing
Author Response
It is an interesting review discussing the reporter culture systems for HEV. The authors discussed the HEV-GLuc Replicon, the recombinant infectious HEV-nanoKAZ, and the recombinant Infectious HEV-HiBiT.
Response: Thank you for your valuable input. We appreciate your interest in our review discussing reporter culture systems for HEV. We have carefully considered your comments and would like to provide the following response:
- Reporter culture system for studying extrahepatic HEV manifestation.
Response: We appreciate your intriguing suggestion regarding the discussion of reporter systems for investigating extrahepatic HEV manifestation. However, it is essential to note that our current manuscript primarily focuses on the evaluation of the HEV-GLuc replicon, the recombinant infectious HEV-nanoKAZ, and the recombinant infectious HEV-HiBiT in the context of drug screening. Therefore, including details about reporter systems for extrahepatic manifestation would deviate from the scope of our review. We will, however, take your suggestions into account for future publications where this specific aspect can be adequately addressed.
- Reporter HEV that used for in vivo animal model.
Response: Thank you for your thoughtful suggestion. Similar to our response to the previous comment, the descriptions about reporter HEV for in vivo animal model will be out of the scope of this current review on the utility of HEV reporter systems within the context of drug screening. Consequently, discussing reporter HEV for in vivo animal models would diverge from our manuscript’s intended focus. We acknowledge the importance of this topic and plan to explore it in depth in future publications.

Reviewer 4 Report
Overview and general recommendation: This review discussed three newly developed HEV reporter systems, which will benefit drug screening against HEV infection. Importantly, the authors also compared their three distinct systems with other existing HEV reporter systems. Overall, the manuscript is very well-written. I only have minor comments below.
Specific points:
1. Table 1 and line 153: Recently, a rabbit HEV-3ra indicator replicon harboring Gaussia luciferase has been developed and tested (PMID: 36809085).
2. Regarding the HEV-nanoKAZ system, is there any effect on HEV replication efficiency and infectivity when a 171-amino acid sequence is inserted in the HVR?
3. Lines 343-369: This part introduces the generation of the HEV-HiBiT and the selection of the linker, which could be shortened, since the details have already been described in the recently published research paper (PMID: 37681960).
4. Curiously, will any of these HEV reporter systems be applied to in vivo studies for anti-HEV drug screening in the future?
Author Response
Overview and general recommendation: This review discussed three newly developed HEV reporter systems, which will benefit drug screening against HEV infection. Importantly, the authors also compared their three distinct systems with other existing HEV reporter systems. Overall, the manuscript is very well-written. I only have minor comments below.
Response: Thank you for your favorable evaluation and insightful suggestions to improve our manuscript. Our point-to-point responses to your specific comments are presented below:
- Table 1 and line 153: Recently, a rabbit HEV-3ra indicator replicon harboring Gaussia luciferase has been developed and tested (PMID: 36809085).
Response: Thank you for your comment. Accordingly, we added the rabbit HEV-3ra indicator replicon harboring Gaussia luciferase to Table 1 and line 220.
- Regarding the HEV-nanoKAZ system, is there any effect on HEV replication efficiency and infectivity when a 171-amino acid sequence is inserted in the HVR?
Response: The nanoKAZ insertion indeed affected the HEV replication efficiency as described in reference no. 61 (Primadharsini et al., J Virol 2022 Fig. 1B). Notably, the HEV RNA titer in PLC/PRF/5 cells transfected with RNA transcripts of pJE03-1760F/P10-nanoKAZ was observed to be lower in comparison to that in the cells transfected with RNA transcripts of the parental clone (pJE03-1760F/P10). It is important to note that while the insertion did affect HEV replication efficiency, it did not result in any significant alteration in infectivity as presented in reference no. 61 (Primadharsini et al., J Virol 2022 Figs. 3C and D).
- Lines 343-369: This part introduces the generation of the HEV-HiBiT and the selection of the linker, which could be shortened, since the details have already been described in the recently published research paper (PMID: 37681960).
Response: Thank you for your thoughtful comment. The data presented in this part have not yet been reported in Nagashima et al., J Virol 2023 [62], and therefore, were described in detail in our current manuscript.
- Curiously, will any of these HEV reporter systems be applied to in vivo studies for anti-HEV drug screening in the future?
Response: We will take your suggestions into account for in depth exploration in future publications.

Round 2
Reviewer 3 Report
No further comments
Moderate language editing